# Prevalence and Molecular Epidemiology of Human Coronaviruses in Africa Prior to the SARS-CoV-2 Outbreak: A Systematic Review

**DOI:** 10.3390/v15112146

**Published:** 2023-10-25

**Authors:** Lisa Arrah Mbang Tambe, Phindulo Mathobo, Mukhethwa Munzhedzi, Pascal Obong Bessong, Lufuno Grace Mavhandu-Ramarumo

**Affiliations:** 1HIV/AIDS & Global Health Research Programme, Faculty of Science, Engineering and Agriculture, University of Venda, Thohoyandou 0950, South Africa; lisa.tambe@univen.ac.za (L.A.M.T.); phindulo.mathobo@univen.ac.za (P.M.); mukhethwa.munzhedzi@gmail.com (M.M.); pascal.bessong@univen.ac.za (P.O.B.); 2Department of Biochemistry and Microbiology, Faculty of Science, Engineering and Agriculture, University of Venda, Thohoyandou 0950, South Africa; 3Centre for Global Health Equity, School of Medicine, 1400 University Ave, Charlottesville, VA 22903, USA

**Keywords:** HCoVs, prevalence, molecular epidemiology, Africa, pandemic preparedness

## Abstract

Coronaviruses, re-emerging in human populations, cause mild or severe acute respiratory diseases, and occasionally epidemics. This study systematically reviewed human coronavirus (HCoVs) infections in Africa prior to the SARS-CoV-2 outbreak. Forty studies on the prevalence or molecular epidemiology of HCoVs were available from 13/54 African countries (24%). The first published data on HCoV was from South Africa in 2008. Eight studies (20%) reported on HCoV molecular epidemiology. Endemic HCoV prevalence ranged from 0.0% to 18.2%. The prevalence of zoonotic MERS-CoV ranged from 0.0% to 83.5%. Two studies investigated SARS-CoV infection, for which a prevalence of 0.0% was reported. There was heterogeneity in the type of tests used in determining HCoV prevalence. Two studies reported that risk factors for HCoV include exposure to infected animals or humans. The quantity of virologic investigations on HCoV on the African continent was scant, and Africa was not prepared for SARS-CoV-2.

## 1. Introduction

Acute respiratory infections (ARIs), including infections with human coronaviruses (HCoV), are the leading cause of morbidity and mortality worldwide. Coronaviruses (CoVs) are enveloped, linear, non-segmented positive-sense single-stranded RNA viruses belonging to the *Coronaviridae* family. They infect both animals and humans [1]. Coronaviruses have one of the largest RNA genomes, ranging from 27–33 kilobases (kb), and are classified into four genera: *Alphacoronavirus*, *Betacoronavirus*, *Gammacoronavirus*, and *Deltacoronavirus* [2,3]. To date, seven HCoV have been described. They fall within the *Alphacoronavirus* (HCoV-NL63 and HCoV-229E) and *Betacoronavirus* (HCoV-OC43, HCoV-HKU1, severe acute respiratory syndrome coronavirus; SARS-CoV, Middle East respiratory syndrome coronavirus; MERS-CoV, and SARS-CoV-2) genera. Endemic HCoVs (HKU1, OC43, NL63, and 229E) occur seasonally, causing mild upper respiratory tract infections in healthy individuals [4], but could also lead to more detrimental lower respiratory tract infections in infants, young children, immunocompromised individuals, persons with comorbidities, and the elderly [5,6,7,8]. The more pathogenic HCoVs (SARS-CoV, MERS-CoV, and SARS-CoV-2) were introduced into the human population through spillover from animals and were responsible for localized epidemics in China [9], the Middle East [10,11], and most recently, the global 2019 coronavirus disease (COVID-19), respectively. These zoonotic HCoVs (SARS-CoV, MERS-CoV, and SARS-CoV-2) lead to more severe disease, compared to the endemic HCoV types.

The continuous re-introduction of HCoVs in the human population over the last three decades has heightened the necessity for surveillance of these pathogens. Prior to the COVID-19 pandemic, most studies investigating the distribution and prevalence of HCoVs were done in regions of the world where the SARS and MERS epidemics were localized. Studies from these regions contributed significantly to improving knowledge on the genetic characteristics, phylogeography, and evolutionary patterns of both endemic and zoonotic HCoVs. To date, molecular epidemiology studies have characterized HCoV-OC43 genotypes (A–K), commonly circulating globally [12,13,14,15,16,17]. Similar genomic investigations have shown that HCoV-NL63 has three main genotypes (A, B, and C), which are common worldwide [18], and further classified into six sub-genotypes (A1–A3 and C1–C3), with sub-genotype C3 being the most recently discovered in paediatric patients in China [19]. For HCoV-229E, it has shown continuous genetic drift over time (Genogroup 1–4), with recent findings identifying two novel genogroups (Genogroups 5 and 6), detected in a COVID-19 patient co-infected with HCoV-229E in Hong Kong [20]. HCoV-HKU1 has three genotypes (A, B, and C), classified based on phylogenetic analysis of the RNA-dependent RNA polymerase (RdRp), Spike (S), and Nucleocapsid (N) genes [21]. These genotypes and sub-genotypes are known to arise due to continuous nucleotide substitution and homologous recombination between circulating strains, which are common events in the Coronaviridae family [22,23].

Understanding the prevalence and molecular epidemiology of HCoVs can contribute to HCoV prediction and control of infection among populations. This systematic review aims to describe the prevalence and molecular epidemiology of HCoVs in Africa prior to the SAR-CoV-2 outbreak.

## 2. Materials and Methods

### 2.1. Search Strategy

The Preferred Reporting Items for Systematic Reviews and Meta-Analyses (PRISMA) approach was used. An electronic search was carried out to identify studies that had reported on HCoV occurrence in Africa prior to the SARS-CoV-2 pandemic. PubMed, Web of Science, and Google Scholar databases were used to search articles from 1 January 1966, when the first HCoV was reported, until 2019. Articles were searched for all three databases using the following search strategy. For PubMed: seroprevalence OR seroepidemiology OR “sero-epidemiology” OR seropositivity OR “sero-epidemiologic studies” OR epidemiology OR prevalence OR incidence OR distribution AND “human coronavirus*” OR “human coronavirus 229E” OR “human coronavirus OC43” OR “human coronavirus NL63” OR “human coronavirus HKU1” OR “severe acute respiratory syndrome coronavirus” OR “middle east respiratory syndrome coronavirus” AND “African country.” Web of Science: seroprevalence OR seroepidemiology OR “sero-epidemiology” OR seropositivity OR “sero-epidemiologic studies” OR epidemiology OR prevalence OR incidence OR distribution AND “human coronavirus*” OR “human coronavirus 229E” OR “human coronavirus OC43” OR “human coronavirus NL63” OR “human coronavirus HKU1” OR “severe acute respiratory syndrome coronavirus” OR “middle east respiratory syndrome coronavirus” AND “African country.” Google Scholar: seroprevalence OR seroepidemiology OR “sero-epidemiology” OR seropositivity OR “sero-epidemiologic studies” OR epidemiology OR prevalence OR incidence OR distribution AND “human coronavirus*” OR “human coronavirus 229E” OR “human coronavirus OC43” OR “human coronavirus NL63” OR “human coronavirus HKU1” OR “severe acute respiratory syndrome coronavirus” OR “middle east respiratory syndrome coronavirus” AND “African country”.

### 2.2. Inclusion and Exclusion Criteria

Published full-text studies and case reports on human coronavirus occurrence and distribution in African countries were selected for examination of their relevance. Articles included in the study met the following criteria: studies published on samples collected prior to 2019, studies that reported on the viral etiology of respiratory viruses (including human coronaviruses) in community and hospital settings; studies that reported on the surveillance, molecular epidemiology, and genomic sequencing of human coronaviruses alone; studies that reported investigation of MERS-CoV in humans and animals; those that reported on retrospective analysis; studies reporting on multiple study sites with samples collected from an African country and a non-African country, and case reports. The following studies were excluded from the analysis: investigation in animals only, studies reporting prevalence of investigated endemic HCoVs (OC43, NL63, HKU1, 229E) based on serology alone, reviews, book chapters, theses, and editorial commentaries. Figure 1 shows the PRISMA flow diagram used in sourcing, identifying, and selecting studies used in the current analysis. Data on the article title and authors, study country, demography, age range study population, year of sample collection, sample size, type of HCoVs detected, method of detection and genotyping, and prevalence for studies that met the inclusion criteria were extracted and collated in Table 1.

## 3. Results

### 3.1. Characteristics of Studies Included in the Analysis

Forty full-text articles met the inclusion criteria and were used for the analysis. Studies that met the inclusion criteria were published between 2008–2021. About 48% (19/40) reported on the prevalence or molecular epidemiology of either endemic HCoVs (OC43, NL63, 229E, HKU1) or zoonotic HCoVs (MERS-CoV, SARS-CoV). About 50% (20/40) of studies sought to determine either the viral etiology, epidemiology, or pattern of occurrence of respiratory viruses. Most studies (62.5%) were conducted in hospital settings, or established influenza-surveillance sentinel sites, where study participants were either admitted, consulting, or receiving vaccination. In 8/40 (20%) studies, investigation was carried out in communities (farms, and households). Two studies (5%) were conducted in an airport setting, while the remaining five studies (12.5%) used a combined approach (hospital and community or hospital and airport).

### 3.2. HCoV Prevalence and Distribution in Africa

The first published data on HCoV was from South Africa in 2008, in which NL63 was described in children less than five years old (Figure 2). Only 13/53 (24%) African countries had data on HCoV prevalence (Table 1) prior to the SARS-CoV-2 outbreak. HCoV prevalence determined through molecular methods was higher (0–95.1%) than that determined by immunofluorescent assays (0–0.18%). The prevalence of endemic HCoVs (OC43, NL63, HKU1, and 229E) ranged between 0.85% in hospitalized children in South Africa to 18.2% in a mixed population (adults and children) at the Grand Magal de Touba in Senegal. Of the 40 studies, 15 of them (36.6%) focused on children alone (0–13 years old), with a reported prevalence ranging from 0.85–10.6% of endemic HCoVs (OC43, NL63, HKU1, and 229E).

In general, the prevalence of MERS-CoV ranged between 0% among Egyptian pilgrims returning from Hajj to 95.1% in a population comprising individuals returning from Saudi Arabia and hospitalized patients in Sudan. Of the 11/40 (27.5%) studies that investigated the occurrence of MERS-CoV, 7/11 (63.6%) of them reported a 0.0% prevalence. These reports investigated MERS-CoV prevalence either in pilgrims returning to their home countries or livestock handlers (including camels), as well as communities without any prior exposure to MERS-CoV. In 3/11 (27.3%) studies, the prevalence ranged from 0.18% in livestock handlers in Garissa and Tana river counties, Kenya, to 83.5% in individuals returning to Sudan from Saudi Arabia and hospital patients. The remaining 1/11 (9.1%) study that investigated MERS-CoV occurrence, was a case report highlighting a family cluster of MERS-CoV in a father and daughter returning to Tunisia from Qatar.

Only 2/40 (5%) studies (one each from Kenya and Sudan) investigated SARS-CoV infection, for which a prevalence of 0.0% was reported.

Regionally, the prevalence of HCoVs across the continent was as follows: Southern Africa (0.85–10.6%), Central Africa (5.3–6.5%), West Africa (0–84.3%), East Africa (0–10%), and North Africa (0–95.1%). Of the 13 countries with published data on HCoV occurrence, Kenya had the highest number of studies published (32.5%), followed by South Africa (15%). This was followed by Senegal, which comprised of 10% of retrieved studies. Both Ghana and Sudan had reports pertaining to 7.5% of all published studies retrieved, while Madagascar, Cote D’Ivoire, and Cameroon had a prevalence of 5% each. The least published data (2.5% each) was from Egypt, Gabon, Nigeria, Tunisia, and Niger.

The proportions of published studies per African region were as follows, in decreasing order: East Africa (15/40; 37.5), West Africa (11/40; 27.5%), Southern Africa (6/40; 15%), North Africa (5/40; 12.5%), and Central Africa (3/40; 7.5%). Figure 3 represents the proportions of published data, while Figure 4 depicts the geographical distribution of studies reporting the occurrence of non-SARS-CoV-2 HCoVs in Africa and testing method used for investigation.

### 3.3. Methodologies for HCoVs Detection

Different detection approaches were employed to determine the prevalence of HCoVs in Africa prior to the SARS-CoV-2 outbreak. These included molecular methods, immunofluorescence assays (IFA), and culture (Table 1). Molecular techniques were used in 35/40 (87.5%) of the studies analysed. Molecular techniques included reverse transcription polymerase chain reaction (RT-PCR), real-time reverse transcription polymerase chain reaction (RT-qPCR), multiplex real-time reverse transcription polymerase chain reaction (mRT-qPCR), and TaqMan array card (TAC) method. These molecular techniques were mostly applied for investigation of endemic HCoVs (70%). These methods were also applied in 5/40 (12.5%) studies investigating zoonotic HCoVs only, and 2/40 (5%) investigating both endemic and zoonotic HCoVs. In 2/40 (5%) studies conducted in Sudan, mRT-qPCR was used with a pancoronavirus panel which simultaneously detects all CoVs (both human and animal), excluding SARS-CoV and MERS-CoV. In one study (2.5%), mRT-qPCR and culture methods were used, and a higher sensitivity was reported for mRT-qPCR compared to culture.

Serological assays such as ELISA, plaque-reduction neutralization test (PRNT), and pseudoparticle neutralization assay (ppNT) were used in 4/40 (10%) studies for the detection of zoonotic MERS-CoV only.

### 3.4. Molecular Epidemiology of HCoVs in Africa Prior to the SARS-CoV-2 Outbreak

Using sequencing, 8/40 (20%) studies reported HCoVs molecular epidemiology; however, only two clearly stated the sequencing method applied (Next Generation Sequencing) (Table 1). Of these eight, 4/8 studies were from Kenya (50%), 1/8 from Ghana (12.5%), 1/8 from South Africa (12.5%), 1/8 from Sudan (12.5%), and 1/8 from Tunisia (12.5%). Findings reported from Kenya described the molecular characteristics of endemic HCoVs between 2008–2018. Three of the four studies (75%) in Kenya described endemic HCoVs in one rural region alone (Kilifi County). They reported the presence of both genotypes of HCoV-NL63 (genotype A and B) circulating in Kilifi county, while genotypes G and H of HCoV-OC43 were most dominant in the population. Genotypes of HCoVs 229E and HKU1 were not reported in this region. The remaining study (1/4) was conducted in the Central, Northern, Western, Highlands, and Coastal regions across Kenya. In this study, they also reported similarity between their sequenced endemic HCoV strains with reference sequences; however, genotypes were not reported (Figure 5).

The studies from Ghana and South Africa reported on genetic characteristics of endemic HCoVs, while those from Sudan and Tunisia reported on MERS-CoV. Studies done on samples collected between 2011–2012 in rural Ghana showed no difference between their endemic HCoV strains and reference sequences. From Cape Town, South Africa, studies done on samples collected between 2004–2005, reported the occurrence of genotype A and B of HCoV-NL63. The study from Sudan reported that samples collected between 2014–2017 from individuals returning from Saudi Arabia and hospital patients showed similarity to MERS-CoV reference sequences from Saudi Arabia and Thailand, respectively. In Tunisia, samples collected in 2014 clustered phylogenetically with geographically diverse MERS-CoV references from Saudi Arabia and the United Arab Emirates.

### 3.5. Risk Factors Associated with HCoV Infection

Only 2/40 studies (5%) investigated risk factors associated with HCoV infection. Both studies conducted in Côte D’Ivoire and Nigeria administered questionnaires to the study participants or cases to ascertain the potential exposure to pathogens. While investigating an outbreak of acute respiratory disease in Côte D’Ivoire, data about associated risk factors, such as exposure to infected animals, persons (living or dead), travel history, and sources of food and water, were collected. They found no link between the source of exposure and the mode of disease transmission. The study conducted in Nigeria investigated the link between occupational exposure (direct or indirect contact) to dromedary camels and infection with MERS-CoV. None of the study participants were infected with MERS-CoV, although they were exposed to MERS-infected dromedary camels.

## 4. Discussion

Prior to the outbreak of SARS-CoV-2, information about the HCoV occurrence, distribution, and prevalence in Africa was sparse. However, post COVID-19, the necessity for continuous HCoVs surveillance has been demonstrated. Thus, strengthening surveillance efforts, implementing standardized testing protocols, provision of required infrastructure, and training of personnel are essential for pandemic preparedness.

While endemic HCoVs (OC43, NL63, 229E, and HKU1) primarily result in mild infections in immune-competent individuals, they are known to contribute to lower respiratory tract infections (LRTIs) in immunocompromised individuals, children ≤ 5 years old, and the elderly, leading to increased mortality [64]. Prior to the outbreak of SARS-CoV-2, studies published in Africa between 2008–2021 reported the occurrence of HCoVs using samples collected between February 2000–December 2019. The current analysis showed that the prevalence of endemic HCoVs (OC43, NL63, 229E, and HKU1) across the continent was between 0.85–18.2% prior to the outbreak of SARS-CoV-2. This may be an underestimation, since most reports (62.5%) were based on hospital setting investigations focused on children ≤ 5 years old. This demographic is known to carry the burden of disease and are prone to ARIs, including infection with endemic HCoVs. Contrarily, immunocompetent individuals ≥ 14 years old are known to have mild or asymptomatic HCoV infections, which mostly go undiagnosed. Thus, near approximate estimates of endemic HCoV prevalence in a population may be unknown. To improve prevalence estimation of endemic HCoV, including community-based studies, such as those conducted on farms, in study cohorts, and during community events, will be beneficial, since it will accommodate symptomatic and asymptomatic individuals (adults and children). This was seen in one cohort survey conducted in Senegal [52], which showed a higher prevalence (18.2%) of endemic HCoVs in the population (8 months–75 years old), compared to what was reported in hospital settings (0.85–10%) of other African regions. Using such community-based approaches could be beneficial in contributing to downstream molecular epidemiology studies, to characterize the genotypes occurring in the population, and potentially contribute to improving diagnostic assay development efforts. A higher prevalence of the zoonotic MERS-CoV (83.5%) was observed in Sudan among a population of returning pilgrims and hospitalized patients [60]. This high prevalence of MERS-CoV may have resulted from high transmission that may have occurred during the Hajj festival among pilgrims while in Saudi Arabia, and later detected upon arrival in Sudan. Such patterns of travelling and large gatherings were also implicated in increasing transmission and spread of variants across the world [65] during the COVID-19 pandemic. Thus, such high prevalence should have alerted Sudanese public health authorities to establish surveillance systems, since most Sudanese will likely travel for Hajj pilgrimage to a MERS endemic area yearly. Similar prevalence of endemic HCoVs (0.2–18.4%) was reported in one review investigating the global seasonality of HCoVs [66]. Of the 22 studies included in their analysis, the majority were conducted in Asia (14 studies), and the least amount in Africa (1 study). Like our study, the reported prevalence was based primarily on patients (adults and children) in hospital settings, presenting with acute respiratory infections (ARIs). This study highlights the dearth of information on endemic HCoVs in the continent, while also highlighting the global need for more non-hospital-based investigations and to gauge prevalence in asymptomatic populations, as well as the circulating genotypes.

Post COVID-19, there is much discussion on pandemic preparedness. Some lessons on effective pandemic preparedness could be taken from Taiwan, which was least affected by the first COVID-19 wave [67]. Taiwan had one of the highest mortality rates due to the SARS epidemic in 2002–2003. As a result of the SARS epidemic, Taiwan set up a surveillance system that was readily deployed in the wake of SARS-CoV-2, and infections were significantly reduced in the first wave of infections, with a moderating effect in subsequent waves.

This review also revealed the paucity of molecular epidemiology studies on HCoVs in Africa prior to the SARS-CoV-2 outbreak. Basic and applied virologic studies are fundamental components for viral pandemic preparations. Through these endeavours, ingredients for the development of detection assays are identified and evaluated; viral genomes are characterized, and epitopes for potential vaccines are identified. Apart from Kenya, where the molecular epidemiology of HCoVs has been continuously investigated, more genomic surveillance studies on HCoVs are needed across Africa as a necessary precursory step for rapid identification of new variants that may arise. This is particularly important since the ease of global human mobility permits silent introductions of new variants across populations. This was evident in studies reported from Sudan and Tunisia, where phylogenetic clustering with MERS-CoV types from Saudi Arabia and UAE was observed in MERS positive patient sequences who returned from the Middle East [60,63]. This phenomenon of travelers introducing HCoV variants into a population was also seen during the COVID-19 pandemic, further emphasizing the need for routine surveillance. Rapid identification of new potentially virulent circulating genotypes allows rapid interception of transmission in the community, thus preventing spread and avoiding epidemics. SARS-CoV was not detected in any of the studies included in the analysis. During the 2002–2003 SARS outbreak, only one case was reported in South Africa [68]. The absence of more cases in Africa during the 2002–2003 SARS outbreak may have been due to two factors. First, the transmissibility of SARS-CoV and MERS-CoV, is reported to be lower than that of SARS-CoV-2. This transmissibility, measured by the basic reproductive rate (R0), is estimated to be 2.4, 0.9, 2.5 for SARS-CoV, MERS-CoV, and SARS-CoV-2, respectively [69,70,71]. Secondly, nosocomial transmission was reported as the main route of infection for SARS-CoV and MERS-CoV cases, since viral shedding peaks during the symptomatic stage of infection. This symptomatic stage, where patients sought medical attention likely increased transmission between patients and healthcare workers [72]. Thus, SARS-CoV may have been transmitted in Africa but this was not detected, even with increased global mobility. Since its eradication in 2003, SARS-CoV has not been detected in the human population.

Third, heterogenous testing methods were applied for HCoVs investigation prior to the outbreak of SARS-CoV-2. Application of molecular techniques was the most common. In terms of pandemic preparedness, this implies the availability of testing methods and facilities. Thus, government research institutions across Africa could pilot and optimize existing protocols in various settings. Through such studies, settings without adequate facilities, necessary infrastructure or equipment, and trained personnel [73] will be identified. While whole genome sequencing (WGS) through next generation sequencing (NGS) reveals aspects of pathogen evolution, diversity, transmission, and spread in a population, more cost-effective methods can be implemented for genomic surveillance, particularly in Africa where resources are limited. Again, lessons could be drawn from the SARS-CoV-2 pandemic in which numerous studies around the world utilized an allele-specific genotyping (ASG) approach for genomic surveillance [74,75,76]. This method was accurate and cost-efficient in variant detection, and could be standardized across the continent for HCoVs monitoring on a larger scale.

Finally, we observed that data on risk factors associated with HCoV infection was scarce. Both studies investigating risk factors reported no zoonotic transmission to humans. Africa hosts a vastly diversified wildlife, bat, and domestic livestock population which harbour diverse coronavirus species [77]. Bats are known hosts of SARS-CoV and SARS-CoV-2, while MERS-CoV is ubiquitous in dromedary camels; both animals are implicated hosts that caused zoonotic spillover to humans. In vivo, in vitro, and ex vivo studies investigating the reason for minimal viral transmission in Africa, even with constant exposure to infected livestock, observed a lower transmission potential in the MERS-CoV strain common in Africa (Clade C), compared to the Arabian Clade A and B strains [78,79,80]. However, continuous phenotypic and molecular epidemiology studies are necessary to monitor any changes that may occur, particularly with the continuous livestock trade between Africa and the Middle East. Livestock with Arabian MERS-CoV strains must be contained to prevent spread, since these strains may outcompete the African Clade C strains, leading to increased zoonotic transmission to occupational workers. Such transmission may rapidly spread in households and communities, which could cause another epidemic.

## 5. Conclusions

In conclusion, this systematic review highlights the dearth in HCoVs investigations in Africa prior to the SARS-CoV-2 pandemic. Hopefully, the SARS-CoV-2 pandemic serves as a wake-up call for the establishment of surveillance systems to monitor HCoVs species in both human and animal African populations. While the majority of Africa is resource-limited, investing in cheaper means of surveillance through wastewater-based methods could be economically beneficial, since it caters for both symptomatic and asymptomatic populations [81,82,83,84,85]. This could be used alongside allele-specific genotyping for sentinel surveillance in households. Establishing and or updating the existing surveillance methods for prevalence and molecular epidemiology of HCoVs will enhance Africa’s contribution to development of diagnostic tests, as well as contribute towards pandemic preparedness.

## Figures and Tables

**Figure 1 viruses-15-02146-f001:**
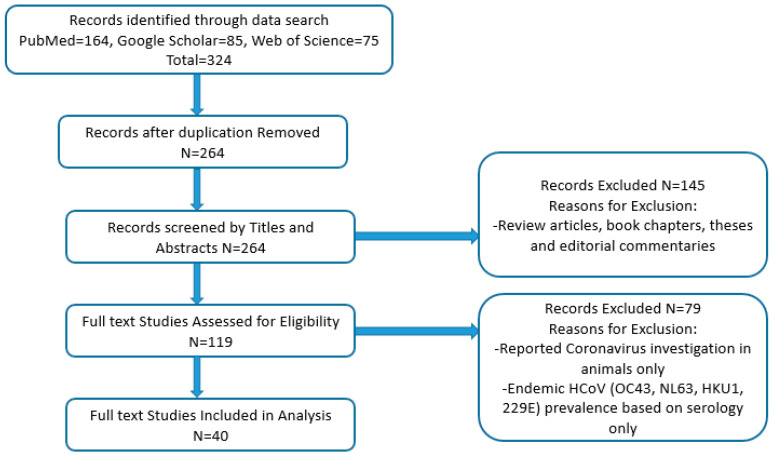
PRISMA flowchart on the screening and selection of studies used for analysis.

**Figure 2 viruses-15-02146-f002:**
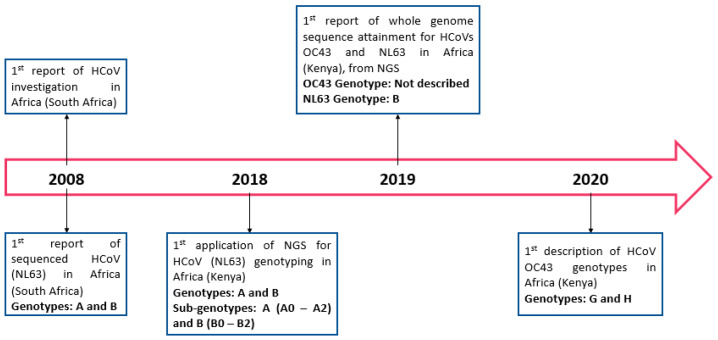
Timeline of studies on HCoVs in Africa prior to the SARS-CoV-2 outbreak.

**Figure 3 viruses-15-02146-f003:**
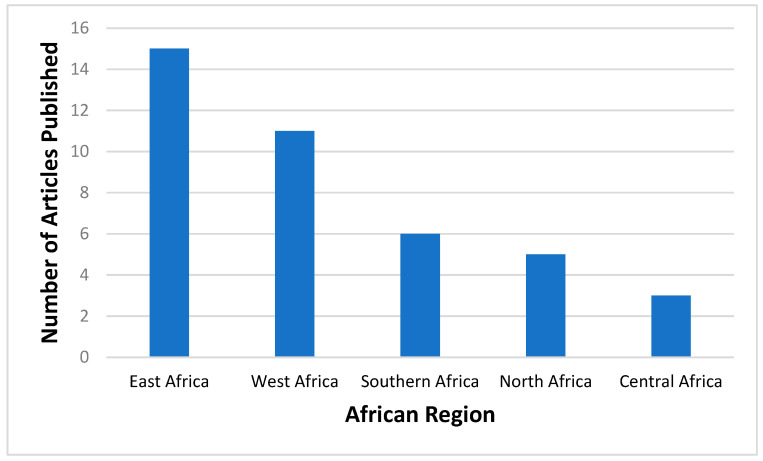
Number of articles published on non-SARS-CoV-2 HCoVs according to different African regions prior to the SARS-CoV-2 outbreak. **Southern Africa-** Botswana, Eswatini, Lesotho, Namibia, South Africa, Zimbabwe; **Central Africa-** Angola, Cameroon, Central Africa Republic, Chad, Congo, Gabon, Democratic Republic of Congo, Equatorial Guinea, Sao Tome and Principe; **West Africa-** Benin, Burkina Faso, Cabo Verde, Cote D’Ivoire, Gambia, Ghana, Guinea, Guinea-Bissau, Liberia, Mali, Mauritania, Niger, Nigeria, Senegal, Sierra Leonne, Togo; **East Africa-** Burundi, Comoros, Djibouti, Eritrea, Ethiopia, Kenya, Madagascar, Malawi, Mauritius, Mozambique, Rwanda, Seychelles, Somalia, South Sudan, Tanzania, Uganda, Zambia; and **North Africa-** Algeria, Egypt, Libya, Morocco, Sudan, Tunisia.

**Figure 4 viruses-15-02146-f004:**
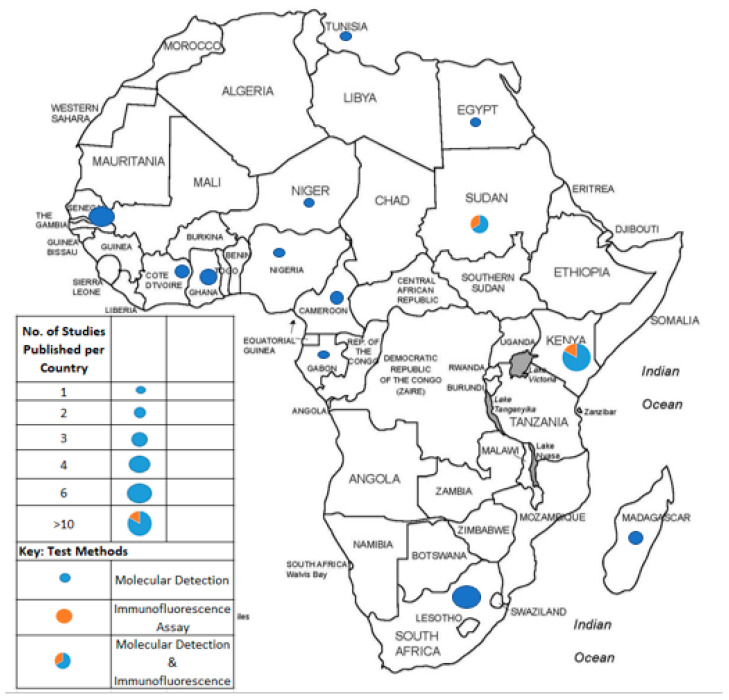
African countries from which studies on non-SARS-CoV-2 HCoV had been published prior to the SARS-CoV-2 outbreak and testing methods applied for investigation.

**Figure 5 viruses-15-02146-f005:**
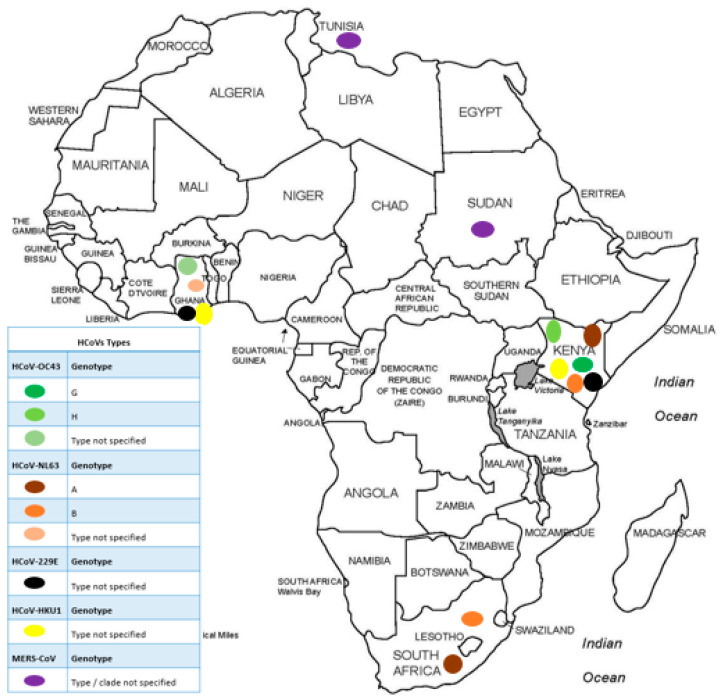
Distribution of HCoVs genotypes in Africa prior to the SARS-CoV-2 outbreak.

**Table 1 viruses-15-02146-t001:** Studies included in the analysis that reported on the prevalence and molecular epidemiology of HCoVs in Africa prior to the SARS-CoV-2 outbreak.

Article Title and Reference	Country	Demography	Age Range	Year of Sample Collection	Sample Size	Type of HCoV Investigated	Method of Detection or Genotyping	HCoV Prevalence	Genotypic Characterization
Viral etiology of severe acute respiratory infections in hospitalized children in Cameroon, 2011–2013 [24]	Cameroon	Children	0–15 years	September 2011–September 2013	347	OC43, 229E, NL63, HKU1	Multiplex RT-qPCR	5.8%	Not investigated
Viral etiology of influenza-like illnesses in Cameroon, January-December 2009 [25]	Cameroon	Adults and Children	1.2 months–75 years	January–December 2009	561	OC43, 229E, NL63, HKU1	One-Step RT-qPCR, Multiplex conventional RT-PCR	5.3%	Not investigated
Detection of non-influenza Viruses in acute respiratory infections in children under 5 years old in Cote D’ivoire (January–December 2013) [26]	Côte D’Ivoire	Children	<5years	January–December 2013	1059	229E, OC43	Multiplex conventional RT-PCR	3.7%	Not investigated
Investigation of an outbreak of acute respiratory disease in Cote D’ivoire in April 2007 [27].	Côte D’Ivoire	Animals, Adults, and Children	0–15+ years	December 2006–February 2007	104	OC43 and 229E	Multiplex conventional RT-PCR, sequencing (method not specified)	1.9%	Sequenced amplified HCoV-OC43 product (results not mentioned)
Cross-sectional survey and surveillance for influenza viruses and MERS-CoV among Egyptian pilgrims returning from Hajj during 2012–2015 [28].	Egypt	Adults and Children	0–105 years	2012–2015	3364	MERS-CoV	RT-qPCR	0%	Not investigated
Viral etiology and seasonality of influenza-like illness in Gabon, March 2010 to June 2011 [29]	Gabon	Adults and Children	10 days–82 years	March 2010–June 2011	1041	NL63, HKU1, 229E, OC43	Multiplex RT-qPCR	6.5%	Not investigated
Human coronaviruses associated with upper respiratory tract infections in three rural areas of Ghana [30]	Ghana	Adults and Children	10+ years	September 2011–September 2012	1213	229E, HKU1, NL63, OC43, MERS-CoV	RT-qPCR and Sequencing (method not specified)	12.4% (MERS-CoV not detected)	Similarity between sequenced HCoV strains and refence sequences
High prevalence of common respiratory viruses and no evidence of Middle East respiratory syndrome coronavirus in Hajj pilgrims returning to Ghana, 2013 [31]	Ghana	Adults	21–85 years	November 2013	839	MERS-CoV	RT-qPCR	0%	Not investigated
Similar virus spectra and seasonality in paediatric patients with acute respiratory disease, Ghana and Germany [32]	Ghana and German Children	Children	0–13 years	February 2008–February 2009	1174	229E, NL63, OC43, HKU1	One-Step RT-qPCR	6.7%	Not investigated
Continuous invasion by respiratory viruses observed in rural households during a respiratory syncytial virus seasonal outbreak in coastal Kenya [33]	Kenya	Adults and Children	4–37 years	December 2009–June 2010	16,928 samples	OC43, NL63, 229E	Multiplex RT-qPCR	7.5%	Not investigated
Comparison of respiratory pathogen yields from nasopharyngeal/oropharyngeal swabs and sputum specimens collected from hospitalized adults in rural Western Kenya [34]	Kenya	Adults	18–49 years	March 2014–July 2015	294	NL63, OC43, HKU1, 229E	TaqMan Array Card	6.1%	Not investigated
Viral etiology of severe pneumonia among Kenyan infants and children [35].	Kenya	Children	1 day–12 years	January–December 2007	759	229E, OC43, NL63, HKU1	RT-qPCR, Sequencing (method not specified)	10%	Results not mentioned
No serologic evidence of Middle East respiratory syndrome coronavirus infection among camel farmers exposed to highly seropositive camel herds: a household linked study, Kenya, 2013 [36]	Kenya	Animals Adults, and Children	5–90 years	2013	760	MERS-CoV	ELISA and plaque-reduction neutralization test (PRNT)	0%	Not investigated
MERS-CoV antibodies in humans, Africa, 2013–2014 [37]	Kenya	Adults and Children	5–90 years	2013–2014	1122	MERS-CoV	ELISA and plaque-reduction neutralization test (PRNT)	0.18%	Not investigated
Molecular characterization of human coronavirus circulating in Kenya, 2009–2012 [38]	Kenya	Adults and Children	2 months–67 years	January 2009–December 2012	417	NL63, HKU1, 229E, OC43, MERS-CoV, SARS-CoV	RT-qPCR; Cell culture, Conventional RT-PCR, Sanger sequencing	8.4% (MERS-CoV and SARS-CoV not detected)	Sequenced samples clustered with reference strains. OC43 and NL63 viruses were under negative selection, albeit not statistically significant.
Infection patterns of endemic human coronaviruses in rural households in coastal Kenya [39]	Kenya	Adults and Children	4–23.4 (IQR)	December 2009–June 2010	483	OC43, NL63, 229E	Multiplex RT-qPCR	7.5%	Not investigated
Surveillance of respiratory viruses in the outpatient setting in rural coastal Kenya: Baseline Epidemiological Observations [40]	Kenya	Adults and Children	0–100 years	January–December 2016	5647	OC43, NL63, 229E	Multiplex RT-qPCR	6.8%	Not investigated
Transmission and evolutionary dynamics of human coronavirus OC43 strains in coastal Kenya investigated by partial spike sequence analysis, 2015–2016 [41]	Kenya	Adults and Children	0–100 years	December 2015–June 2016	3314	OC43	Multiplex RT-qPCR, Conventional RT-PCR, Sanger sequencing of Spike Gene	2.8%	Sequenced samples clustered with OC43 reference genotypes G (85%) and H (15%)
Surveillance of endemic human coronaviruses (HCoV-NL63, OC43 and 229E) associated with childhood pneumonia in Kilifi, Kenya [42]	Kenya	Children	0–4 years	January 2007–December 2019	7957	NL63, OC43, 229E	Multiplex RT-qPCR	3.9%	Not investigated
Improved detection of respiratory viruses in pediatric outpatients with acute respiratory illness by real-time PCR using nasopharyngeal flocked swabs [43]	Kenya	Adults and Children	0–12 years	January–April 2009	299	OC43, NL63, 229E	Multiplex RT-qPCR	7.4%	Not investigated
Human coronavirus NL63 molecular epidemiology and evolutionary patterns in rural coastal Kenya [44].	Kenya	Adults and Children	0–100 years	February 2008–May 2014	22,491	NL63	RT-PCR, HiSeq NGS	2.1%	NL63 genotype A and B observed, with six lineages (A0–A2 and B0–B2)
Genome sequences of human coronavirus OC43 and NL63, associated with respiratory infections in Kilifi, Kenya [45]	Kenya	Children	2 months–13 years	2017, 2018	3	OC43, NL63	MiSeq NGS	Retrospective genomic study	OC43 genomes clustered in distinct genome-based phylogeny branches. NL63 genomes clustered with genotype B
Viral and atypical bacterial etiology of acute respiratory infections in children under 5 years old living in a rural tropical area of Madagascar [46]	Madagascar	Children	2–59 months	February 2010–February 2011	295	NL63, 229E, OC43, HKU1	Multiplex RT-qPCR	8%	Not investigated
Viral etiology of influenza-like illnesses in Antananarivo, Madagascar, July 2008 to June 2009 [47]	Madagascar	Adults and Children	3 months–77 years	July 2008–June 2009	313	NL63, 229E, OC43, HKU1	Multiplex RT-qPCR, RT-qPCR	9.6%	Not investigated
Molecular detection of respiratory pathogens among children aged younger than 5 years hospitalized with febrile acute respiratory infections: A prospective hospital-based observational study in Niamey, Niger [48]	Niger	Children	0–4 years	January–December 2015	638	OC43, 229E, NL63, HKU1	RT-qPCR	8.0%	Not investigated
Lack of serological evidence of Middle East respiratory syndrome coronavirus infection in virus exposed camel abattoir workers in Nigeria, 2016 [49]	Nigeria	Humans and Animals	Not specified	October 2015–February 2016	311	MERS-CoV	ELISA, pseudoparticle neutralization assay (ppNT)	0%	Not investigated
Influenza-like illnesses in Senegal: not only focus on influenza viruses [50]	Senegal	Adults and Children	0–25+ years	May 2012–June 2013	1427	OC43, 229E, NL63	Multiplex RT-qPCR	2%	Not investigated
Viral etiology of respiratory infections in children under 5 years old living in tropical rural areas of Senegal: The EVIRA project [51]	Senegal	Children	0–4 years	July–December 2007	67	OC43, NL63, 229E, HKU1	Multiplex Conventional RT-PCR	7.3%	Not investigated
Respiratory and gastrointestinal infections at the 2017 Grand Magal de Touba, Senegal: a prospective cohort survey [52]	Senegal	Adults and Children	8 months–75 years	4th–23rd November 2017	123	NL63, 229E, OC43, HKU1	One Step Duplex RT-PCR	18.2%	Not investigated
Respiratory viruses associated with patients older than 50 years presenting with ILI in Senegal, 2009 to 2011 [53]	Senegal	Adults	50–97 years	January 2009–December 2011	232	NL63, 229E, OC43	Two-Step RT-qPCR	2.3%	Not investigated
Human coronavirus NL63 infections in infants hospitalised with acute respiratory tract infections in South Africa [54]	South Africa	Children	13 days–5 years	2003–2004	1055	NL63	Conventional RT-PCR	0.85%	Not investigated
Role of human metapneumovirus, human coronavirus NL63 and human bocavirus in infants and young children with acute wheezing [55]	South Africa	Children	2 months–6 years	May 2004–November 2005	242	NL63	Conventional RT-PCR, Sanger sequencing	2.5%	NL63 genotype A and B detected
Human rhinovirus infection in young African children with acute wheezing [56]	South Africa	Children	2 months–5 years	May 2004–November 2005	220	NL63	Conventional RT-PCR	1.3%	Not investigated
Contribution of common and recently described respiratory viruses to annual hospitalizations in children in South Africa [57]	South Africa	Children	0–4 years	2006–2007	610	NL63, OC43, 229E, HKU1	Multiplex RT-qPCR	4.4%	Not investigated
Clinical epidemiology of bocavirus, rhinovirus, two polyomaviruses and four coronaviruses in HIV-infected and HIV-uninfected South African children [58]	South Africa	Children	1 month–2 years	February 2000 to January 2002	1460	NL63, OC43, 229E, HKU1	Multiplex RT-qPCR	10.6%	Not investigated
Human bocavirus, coronavirus, and polyomavirus detected among patients hospitalised with severe acute respiratory illness in South Africa, 2012 to 2013 [59]	South Africa	Adults and Children	<1–65+ years	January 2012–December 2013	680	NL63, HKU1, OC43, 229E	Multiplex RT-qPCR	4.8%	Not investigated
Detection, identification and sequencing of Middle East respiratory syndrome Coronavirus (MERS-CoV) among Sudanese patients [60]	Sudan	Adults and Children	<20–100 years	2014–2017	200	MERS-CoV, Pancoronavirus (229E, OC43, HKU1, NL63, SARS-CoV)	Conventional One-Step RT-PCR, Sequencing	95.1% (83.5% MERS-CoV; 11.6%Pancoronavirus)	Sequenced MERS-CoV samples from the hospital and airport clustered with strains from Thailand and Saudi Arabia, respectively.
Detection of some respiratory viruses by molecular techniques among two Sudanese targets individual [61]	Sudan	Adults and Children	<20–100 years	2014–2017	200	MERS-CoV and Pancoronavirus (229E, OC43, HKU1, NL63, SARS-CoV)	Conventional One-Step RT-PCR (using Pancoronavirus panel)	95.1% (83.5% MERS-CoV; 11.6%Pancoronavirus)	Not investigated
MERS-CoV in camels but not camel handlers, Sudan, 2015 and 2017 [62]	Sudan and Qatar	Adults and Animals	Not specified	2015–2017	56	MERS-CoV	Spike (S1) protein microarray, S1 protein-based ELISA	0%	Not investigated
Family cluster of Middle East respiratory syndrome coronavirus infections, Tunisia, 2013 [63]	Tunisia	Adults	30–66 years	2013	14	MERS-CoV	RT-qPCR	21%	Sequenced sample clustered with reference sequences from Saudi Arabia and United Arab Emirates.

## Data Availability

All data used in this study is available in the article.

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
