# Peer review of "Prevalence and Molecular Epidemiology of Human Coronaviruses in Africa Prior to the SARS-CoV-2 Outbreak: A Systematic Review"

_viruses, 2023, doi:10.3390/v15112146_

Round 1

Reviewer 1 Report

Comments and Suggestions for Authors

The manuscript provides a systematic review on the coronaviruses in humans in Africa. It is a nice overview and provides information on the level of screening-capacity (molecular assays) in this continent. A major item which would improve the manuscript is to have a clear distinction between the seasonal endemic human coronaviruses (HCoVs) NL63, HKU1, 229E and OC43 on the one hand, and the other coronaviruses MERS-CoV, and SARS-CoV that are not endemic. These latter have not been among us humans for centuries – like the HCoVs – and should thus be treated separately. Prevalence on MERS-CoV and SARS-CoV can be investigated by PCR-based or sequencing-based assays, but also via serology (e.g. ELISAs). Prevalence studies for the 4 HCoVs, can only be based on PCR- or sequencing based nucleic acid detection, as all/most adults are seropositive for these viruses. Inclusion of reference 33 (El Duah et al, 2019) in the paragraph on prevalence is therefore a false (high) prevalence number. A strong advice is thus to not include this study. Serology can be used for MERS-CoV and SARS-CoV studies, as these viruses are not endemic. But it is important also for these viruses to provide the information that the data come from serological studies. Suggestions for improvements are listed below.

-        Abstract (and the whole manuscript). Make a clear distinction between the groups. There is the endemic HCoVs (NL63, 229E, HKU1 and OC43), secondly there is MERS-CoV, which is once in a while still seen in humans nowadays (mainly Saudi Arabia), and SARS-CoV, a virus that has been successfully eliminated from the human population in 2003, and is no longer circulating in humans . Do not merge prevalence’s between these groups, as it leads to non-informative values.

-        Introduction, line 30. Coronaviridae is written in italics

-        Introduction, line 32. The four genera are written in italics

-        Introduction, line 57. HCoVs NL63, OC43 and HKU1 have co-circulating types. But there are no co-circulating genotypes for HCoV-229E. This virus only shows genetic drift. The Hong Kong paper (20, Lau et al 2021) provides the most recent genotype in 2021, and not a co-circulation genotype. In that Hong Kong paper it is mentioned in the correct way (Discussion section): “ Phylogenetic analysis of the S gene showed that HCoV-229E has continued to evolve with time to generate new genogroups, with clustering of strains in chronological order.” That “chronological order” is the clue: different genotype do not co-circulate. It is also not so that the genetic characteristics of HCoV-229E are the least studied. Many studies have characterized HCoV-229E strains, and they all show that there is antigenic drift, each time 229E-viruses evolving from its previous version. Fist study on this was already in 2006 by Chibo and Birch https://pubmed.ncbi.nlm.nih.gov/16603522/ , and many studies since then have further confirmed the link between chronicity (time) and evolution.

-        Table 1: Delete the information of reference 33 as it is based on serology. The 84,3% prevalence for the HCoVs should therefore not be included. As mentioned above, it is as high as expected for sero-prevalence, because it is known that seroconversion occurs already at an early age, with remaining presence for antibodies. See Zhou et al: https://bmcinfectdis.biomedcentral.com/articles/10.1186/1471-2334-13-433, and Dijkman et al  https://pubmed.ncbi.nlm.nih.gov/22188723/

-        Results, line 131 and further. Add information on the type of testing, whether prevalence is based on nucleic acid detection, or serology.

-        Results, line 131 and further. Make the distinction into the three: 1. HCoVs / 2. MERS-CoV /3.  SARS-CoV.

-        Discussion, line 307. In vivo, in vitro, ex vivo is written in italics and not with capitals

-        Discussion, line 277. This sentence may best be deleted. It is not strange that SARS-CoV was not detected. That virus is since 2003 not circulating in humans thanks to massive quarantine-measures in amongst others China and Canada.

Author Response

General Comment: The manuscript provides a systematic review on the coronaviruses in humans in Africa. It is a nice overview and provides information on the level of screening-capacity (molecular assays) in this continent. A major item which would improve the manuscript is to have a clear distinction between the seasonal endemic human coronaviruses (HCoVs) NL63, HKU1, 229E and OC43 on the one hand, and the other coronaviruses MERS-CoV, and SARS-CoV that are not endemic. These latter have not been among us humans for centuries – like the HCoVs – and should thus be treated separately. Prevalence on MERS-CoV and SARS-CoV can be investigated by PCR-based or sequencing-based assays, but also via serology (e.g. ELISAs). Prevalence studies for the 4 HCoVs, can only be based on PCR- or sequencing based nucleic acid detection, as all/most adults are seropositive for these viruses. Inclusion of reference 33 (El Duah et al, 2019) in the paragraph on prevalence is therefore a false (high) prevalence number. A strong advice is thus to not include this study. Serology can be used for MERS-CoV and SARS-CoV studies, as these viruses are not endemic. But it is important also for these viruses to provide the information that the data come from serological studies. Suggestions for improvements are listed below.

Comment 1:  Abstract (and the whole manuscript). Make a clear distinction between the groups. There is the endemic HCoVs (NL63, 229E, HKU1 and OC43), secondly there is MERS-CoV, which is once in a while still seen in humans nowadays (mainly Saudi Arabia), and SARS-CoV, a virus that has been successfully eliminated from the human population in 2003 and is no longer circulating in humans. Do not merge prevalence’s between these groups, as it leads to non-informative values.

Response: This distinction has been made all through the text. This can be seen in the abstract, introduction, and results section.

In the abstract section, line 18 – 19, page 1, was previously written as “HCoV prevalence ranged from 0.0% to 95.1%. The prevalence of MERS-CoV ranged from 0.18% to 83.5%”. This has been amended to “Endemic HCoV prevalence ranged from 0.0% to 18.2%. The prevalence of zoonotic MERS-CoV ranged from 0.0% to 83.5%”.

In the introduction section, line 37 – 46, page 1 – 2, was previously written as “Endemic HCoVs (HKU1, OC43, NL63, and 229E) occur seasonally, causing mild upper respiratory tract infections in healthy individuals [4], but could also lead to more detrimental lower respiratory tract infections in infants, young children, immunocompromised individuals, persons with comorbidities, and the elderly [5–8]. SARS-CoV, MERS-CoV, and SARS-CoV-2 were introduced into the human population through spill-over from animals and were responsible for localized epidemics in China [9], the Middle East [10,11], and most recently the global 2019 coronavirus disease (COVID-19), respectively”. This has been amended as follows: “Endemic HCoVs (HKU1, OC43, NL63, and 229E) occur seasonally, causing mild upper respiratory tract infections in healthy individuals [4], but could also lead to more detrimental lower respiratory tract infections in infants, young children, immunocompromised individuals, persons with comorbidities, and the elderly [5–8]. The more pathogenic HCoVs (SARS-CoV, MERS-CoV, and SARS-CoV-2) were introduced into the human population through spillover from animals and were responsible for localized epidemics in China [9], the Middle East [10,11], and most recently the global 2019 coronavirus disease (COVID-19), respectively. These zoonotic HCoVs (SARS-CoV, MERS-CoV, and SARS-CoV-2) lead to more severe disease compared to the endemic HCoV types”.

In the results section, in lines 120 – 122, this was written as “About 49% (20/41) reported on the prevalence or molecular epidemiology of HCoVs (OC43, NL63, 229E, HKU1, MERS-CoV, SARS-CoV) alone”. This has been amended to “About 48% (19/40) reported on the prevalence or molecular epidemiology of either endemic HCoVs (OC43, NL63, 229E, HKU1) or zoonotic HCoVs, (MERS-CoV, SARS-CoV)”, in lines 129 – 131.

Comment 2: Introduction, line 30. Coronaviridae is written in italics

Response: Coronaviridae has been put in italics in Introduction (line 30).

Comment 3: Introduction, line 32. The four genera are written in italics.

Response: Alphacoronavirus, Betacoronavirus, Gammacoronavirus, and Deltacoronavirus have been put in italics. This can be seen in the introduction, line 32 – 33.

Comment 4: Introduction, line 57. HCoVs NL63, OC43 and HKU1 have co-circulating types. But there are no co-circulating genotypes for HCoV-229E. This virus only shows genetic drift. The Hong Kong paper (20, Lau et al 2021) provides the most recent genotype in 2021, and not a co-circulation genotype. In that Hong Kong paper it is mentioned in the correct way (Discussion section): Phylogenetic analysis of the S gene showed that HCoV-229E has continued to evolve with time to generate new genogroups, with clustering of strains in chronological order.” That “chronological order” is the clue: different genotype do not co-circulate. It is also not so that the genetic characteristics of HCoV-229E are the least studied. Many studies have characterized HCoV-229E strains, and they all show that there is antigenic drift, each time 229E-viruses evolving from its previous version. First study on this was already in 2006 by Chibo and Birch: https://pubmed.ncbi.nlm.nih.gov/16603522/, and many studies since then have further confirmed the link between chronicity (time) and evolution.

Response: This has been corrected. The original statement in the introduction read as: “Compared to the other HCoVs, the genetic characteristics and evolutionary mechanisms of HCoV-229E are the least studied. However, recent findings indicate that there are six HCoV-229E genogroups (Genogroup 1 – 6), with Genogroups 5 and 6, detected recently in a COVID-19 patient co-infected with HCoV-229E in Hong Kong [20]” in lines 55 – 59, has been changed.

This now reads thus: “for HCoV-229E, it has shown continuous genetic drift over time (Genogroup 1 – 4), with recent findings identifying two novel genogroups (Genogroups 5 and 6), detected in a COVID-19 patient co-infected with HCoV-229E in Hong Kong [20]” in lines 58 – 62

Comment 5: Table 1: Delete the information of reference 33 as it is based on serology. The 84,3% prevalence for the HCoVs should therefore not be included. As mentioned above, it is as high as expected for seroprevalence, because it is known that seroconversion occurs already at an early age, with remaining presence for antibodies. See Zhou et al: https://bmcinfectdis.biomedcentral.com/articles/10.1186/1471-2334-13-433, and Dijkman et al https://pubmed.ncbi.nlm.nih.gov/22188723/.

Response: This reference has been deleted in the table and list of references. All reported analysis is now based on 40 studies, and the PRISMA flow diagram has also been updated. Please see page 4, figure 1.

Comment 6: Results, line 131 and further. Add information on the type of testing, whether prevalence is based on nucleic acid detection, or serology.

Response: This has been implemented in the results section.

In the results section, lines 143 – 145, describes the prevalence obtained using the molecular or serology methods. This reads as “HCoV prevalence determined through molecular methods was higher (0 – 95.1%) than that determined by immunofluorescent assays (0 – 0.18%)”.

A distinction between the prevalence of endemic HCoVs (OC43, NL63, 229E, HKU1) and zoonotic HCoVs (MERS-CoV and SARS-CoV) was also highlighted. Please see the reference points below.

Lines 145 – 148: The prevalence of endemic HCoVs (OC43, NL63, HKU1, and 229E) ranged between 0.85% in hospitalized children in South Africa to 18.2% in a mixed population (adults and children) at the Grand Magal de Touba in Senegal.

Lines 154 – 156: In general, the prevalence of MERS-CoV ranged between 0% among Egyptian pilgrims returning from Hajj to 95.1% in a population comprising individuals returning from Saudi Arabia and Hospitalized patients in Sudan.

Lines 165 – 166: Only 2/40 (5%) studies (one each from Kenya and Sudan) investigated SARS-CoV infection for which a prevalence of 0.0% was reported.

Lines 212 – 222: These molecular techniques were mostly applied for investigation of endemic HCoVs (70%). These methods were also applied in 5/40 (12.5%) studies investigating zoonotic HCoVs only, and 2/40 (5%) investigating both endemic and zoonotic HCoVs. In 2/40 (5%) studies conducted in Sudan, mRT-qPCR was used with a pancoronavirus panel which simultaneously detects all CoVs (both human and animal), excluding SARS-CoV and MERS-CoV. In one study (2.5%) mRT-qPCR and culture methods were used, and a higher sensitivity was reported for mRT-qPCR compared to culture.

Serological assays such as ELISA, plaque-reduction neutralization test (PRNT), and pseudoparticle neutralization assay (ppNT) were used in 4/40 (10%) studies for the detection of zoonotic MERS-CoV only.  

Comment 7: Results, line 131 and further. Make the distinction into the three: 1. HCoVs / 2. MERS-CoV /3. SARS-CoV.

Response: This distinction has been made throughout the manuscript:

Results section, lines 129 – 131: About 48% (19/40) reported on the prevalence or molecular epidemiology of either endemic HCoVs (OC43, NL63, 229E, HKU1) or zoonotic HCoVs, (MERS-CoV, SARS-CoV)”.

Results section, lines 145 – 148: The prevalence of endemic HCoVs (OC43, NL63, HKU1, and 229E) ranged between 0.85% in hospitalized children in South Africa to 18.2% in a mixed population (adults and children) at the Grand Magal de Touba in Senegal.

Results section, lines 154 – 156: In general, the prevalence of MERS-CoV ranged between 0% among Egyptian pilgrims returning from Hajj to 95.1% in a population comprising individuals returning from Saudi Arabia and Hospitalized patients in Sudan.

Results section, lines 165 – 166: Only 2/40 (5%) studies (one each from Kenya and Sudan) investigated SARS-CoV infection for which a prevalence of 0.0% was reported.

Comment 8: Discussion, line 307. In vivo, in vitro, ex vivo is written in italics and not with capitals.

Response: In vivo, in vitro and ex vivo have been italicized. Please see page 8, line 375.

Comment 9: Discussion, line 277. This sentence may best be deleted. It is not strange that SARS-CoV was not detected. That virus is since 2003 not circulating in humans thanks to massive quarantine-measures in amongst others China and Canada.

Response: This sentence has been deleted. Correction included reads thus: “SARS-CoV was not detected in any of the studies included in the analysis”, line 343 – 344. During the 2002 – 2003 SARS outbreak, only one case was reported in South Africa [68]. The absence of more cases in Africa during the 2002 – 2003 SARS outbreak may have been due to two factors. First, the transmissibility of SARS-CoV and MERS-CoV, is reported to be lower than that of SARS-CoV-2. This transmissibility, measured by the basic reproductive rate (R0), is estimated to be 2.4, 0.9, 2.5 for SARS-CoV, MERS-CoV, and SARS-CoV-2, respectively [69 – 71]. Secondly, nosocomial transmission was reported as the main route of infection for SARS-CoV and MERS-CoV cases, since viral shedding peaks during the symptomatic stage of infection. This symptomatic stage, where patients sought medical attention likely increased transmission between patients and healthcare workers [72]. Thus, SARS-CoV may have been transmitted in Africa but this was not detected, even with increased global mobility. Since its eradication in 2003, SARS-CoV has not been detected in the human population (lines 344 – 355).

Reviewer 2 Report

Comments and Suggestions for Authors

LAM Tambe and colleagues performed a systematic literature review on the molecular epidemiology of human coronavirus before 2019. The review is thorough, well written, and fills a hole in this under-researched area.

The following suggestions would further improve the manuscript:

1) The discussion would benefit form similar reviews of other parts of the world

2) Three of the studies used in this review have a much higher prevalence (>85%) than the remaining studies. Please discuss potential root causes such as differences in study population, testing methods etc.

3) The majority of studies have been performed in hospital or influenza surveillance settings. Many coronavirus infections are mild or asymptomatic and are likely not detected in these settings. Please discuss how this might affect the results of this review.

4) Figure 4 could be more informative by using different sized circles, where the size of the circle corresponds to, for example, number of studies, number of samples tested, number of samples tested as ratio of total population etc. The color of the circle could be used to indicate different testing methods.

5) For figure 5, consider a different color scheme: different shades of red for OC43 genotypes, different shades of green for NL63,...

Author Response

General Comment: LAM Tambe and colleagues performed a systematic literature review on the molecular epidemiology of human coronavirus before 2019. The review is thorough, well written and fills a hole in this under-researched area. The following suggestions would further improve the manuscript:

Comment 1: [The discussion would benefit from similar reviews of other parts of the world].   

Response: This has been implemented. We compared one of the most recent reviews (Park et al., 2020; reference 66) with the findings in our study. This can be found in page 7, from line 310 – 317, as “Similar prevalence of endemic HCoVs (0.2 – 18.4%) was reported in one review investigating the global seasonality of HCoVs [66]. Of the 22 studies included in their analysis, majority were conducted in Asia (14), and the least in Africa (1). Like our study, the re-ported prevalence was based primarily on patients (adults and children) in hospital set-tings, presenting with acute respiratory infections (ARIs). This study highlights the dearth of information on endemic HCoVs in the continent, while also highlighting the global need for more non-hospital-based investigations, to gauge prevalence in asymptomatic populations, as well as the circulating genotypes.

Comment 2: Three of the studies used in this review have a much higher prevalence (>85%) than the remaining studies. Please discuss potential root causes such as differences in study population, testing methods etc.   

Response: Per recommendations from Reviewer 1, one of the studies (El-Duah et al., 2019; previously reference 33) was removed from the analysis. This is because the high prevalence (84.3%) of HCoV-NL63, an endemic HCoV reported is based on serology. This is expected for seroprevalence, because it is known that seroconversion occurs already at an early age, with remaining presence for antibodies. Thus, this study may give a false representation of the prevalence in the study population, compared to other studies that used more sensitive molecular or immunofluorescence assay techniques.

For the remaining two studies (Ibrahim et al., 2018; reference 60 and Ibrahim et al., 2018; reference 61), whose prevalence is based on zoonotic MERS-CoV, we discussed this higher prevalence (83.5%) in lines 301 – 310, page 7 as follows: “A higher prevalence of the zoonotic MERS-CoV (83.5%) was observed in Sudan among a population of returning pilgrims and hospitalized patients [60]. This high prevalence of MERS-CoV may have resulted from high transmission that may have occurred during the Hajj festival among pilgrims while in Saudi Arabia, and later detected upon arrival in Sudan. Such patterns of travelling and large gatherings were also implicated in increasing transmission and spread of variants across the world [65] during the COVID-19 pandemic. Thus, such high prevalence should have alerted Sudanese public health authorities to establish surveillance systems, since most Sudanese will likely travel for Hajj pilgrimage to a MERS endemic area yearly”.

Comment 3: The majority of studies have been performed in hospital or influenza surveillance settings. Many coronavirus infections are mild or asymptomatic and are likely not detected in these settings. Please discuss how this might affect the results of this review.   

Response: This is discussed in lines 286 – 301, page 7 as follows: “This may be an underestimation, since most reports (62.5%) were based on hospital set-tings investigations, focused on children ≤5years old. This demography is known to carry the burden of disease, and are prone to ARIs, including infection with endemic HCoVs. Contrarily, immunocompetent individuals ≥14years old are known to have mild or asymptomatic HCoV infections, which mostly go undiagnosed. Thus, near approximate estimates of endemic HCoV prevalence in a population may be unknown. To improve prevalence estimation of endemic HCoV, including community-based studies, such as those conducted on farms, in study cohorts, and during community events, will be beneficial, since it will accommodate symptomatic and asymptomatic individuals (adults and children). This was seen in one cohort survey conducted in Senegal [52], which showed a higher prevalence (18.2%) of endemic HCoVs in the population (8 months – 75 years old), compared to what was reported in hospital settings (0.85 – 10%) of other African regions. Using such community-based approaches could be beneficial in contributing to down-stream molecular epidemiology studies, to characterize the genotypes occurring in the population, and potentially contribute to improving diagnostic assay development efforts”.                                                                     

Comment 4: Figure 4 could be more informative by using different sized circles, where the size of the circle corresponds to, for example, number of studies, number of samples tested, number of samples tested as ratio of total population etc. The color of the circle could be used to indicate different testing methods.   

Response: This has been modified to emphasize two points. First, the size of the circle corresponding to the number of studies and different colors were used to indicate different testing methods. Please see Figure 4, page 3.

Comment 5: For figure 5, consider a different color scheme: different shades of red for OC43 genotypes, different shades of green for NL63,..   

Response: We have modified Figure 5 as recommended. Please see Figure 5, page 6.

Round 2

Reviewer 1 Report

Comments and Suggestions for Authors

All suggestions for improvement have been incorporated